# Sustainable Floodplains: Linking E-Flows to Floodplain Management, Ecosystems, and Livelihoods in the Sahel of North Africa

**Gordon C. O'Brien [1],\*, Chris Dickens [2] , Chris Baker [3], Retha Stassen [1] and Frank van Weert [3]**

[1] School of Biology and Environmental Sciences, Faculty of Agriculture and Natural Sciences, University of Mpumalanga, Private Bag X11283, Nelspruit 1200, South Africa; rethas@lantic.net

[2] International Water Management Institute–Colombo, 127, Sunil Mawatha, Battaramulla, Colombo 10120, Sri Lanka; c.dickens@cgiar.org

[3] Wetlands International, P.O. Box 471, 6700 AL Wageningen, The Netherlands; Chris.Baker@wetlands.org (C.B.); Frank.vanWeert@wetlands.org (F.v.W.)

\* Correspondence: gordon.obrien@ump.ac.za

**Abstract:** Floodplains are particularly important in the semi-arid region of the Sub-Sahelian Africa. In this region, water governance is still being developed, often without adequate information and technical capacity for good, sustainable water resource management. However, water resources are being allocated for use with minimal sustainability considerations. Environmental flows (e-flows) include the quantity and timing of flows or water levels needed to meet the sustainable requirements of freshwater and estuarine ecosystems. Holistic regional scale e-flows linked to floodplain management can make a noticeable contribution to sustainable floodplain management. The Inner Niger Delta (IND) in Mali is an example of a vulnerable, socio-ecologically important floodplain in the Sahel region of North Africa that is being developed with little understanding of sustainability requirements. Although integrally linked to the Upper Niger River catchment, the IND sustains a million and half people within the region and exports food to surrounding areas. The flooding of the Delta is the engine of the socio-economic development as well as its ecological integrity. This paper aims to demonstrate the contribution that holistic regional e-flow assessment using the PROBFLO approach has to achieving floodplain sustainability. This can be achieved through the determining the e-flow requirements to maintain critical requirements of the ecosystems and associated services used by local vulnerable human communities for subsistence and describing the socio-ecological consequences of altered flows. These outcomes can contribute to the management of the IND. In this study, the socio-ecological consequences of altered flows have been evaluated by assessing the risk of alterations in the volume, duration, and timing of flows, to a number of ecological and social endpoints. Based on the risk posed to these endpoints by each scenario of change, an e-flow of 58% (26,685 million cubic meters (MCM) of water annually) was determined that would protect the ecosystem and maintain indicator components at a sustainable level. These e-flows also provide sustainable services to local communities including products for subsistence and limit any abnormal increases in diseases to the vulnerable African communities who live in the basin. Relative risk outputs for the development scenarios result in low-to-high-risk probabilities for most endpoints. The future development scenarios include insufficient flows to maintain sustainability during dry or low-flow periods with an increase in zero flow possibilities. Although unsuitable during the low-flow or dry periods, sufficient water is available through storage in the basin to meet the e-flows if these scenarios were considered for implementation. The IND is more vulnerable to changes in flows compared to the rivers upstream of the IND. The e-flow outcomes and consequences of altered flow scenarios has contributed to the management of vulnerable IND floodplains and the requirements and trade-off considerations to achieve sustainability.

**Keywords:** floodplains; sustainability; environmental flows; Inner Niger Delta; social and ecological endpoints; PROBFLO

## 1. Introduction

Floodplains are areas of land adjacent to a stream or river that stretches from the banks of the river channel to the base of the enclosing valley walls, where flooding occurs during periods of high flows [1]. Across the globe, floodplains form key wetland habitats that sustain high biodiversity and socio-ecological processes that support the livelihoods of some of our most vulnerable human communities. These communities often depend on the crops, grazing, abundant fish, and other natural products that healthy fertile floodplain systems provide. It is assumed that of the 850,000 km$^2$ of Sub-Saharan African (SSA) floodplains [2] about 250,000 km$^2$ is used as farming land sustaining more than 50 million, often smallholder farmers [3].

These floodplain systems are of particular importance in the semi-arid region of the SSA such as the Inner Niger Delta (IND) in Mali, the Sudd in South Sudan, and the rivers and lakes of the Tchad Basin. Rains feeding the rivers in the more humid tropical regions to the south drive seasonal flooding of the floodplain systems after which the floods recede leaving lakes, marshes, and exposed productive land behind. The floodplains comprise the river channels and permanent water bodies such as lakes and wetlands creating a heterogeneous mosaic of ecosystem habitats each providing essential provisioning ecosystem services for these livelihoods. Additional services include reduced natural hazards from unwanted extreme floods through flood attenuation, sequestered carbon in the permanently wet areas, and improved water quality through purifying biogeochemical processes and dilution of diffuse and point source pollution. Over thousands of years, communities have developed their livelihood strategies to capitalize on the bounty that the floodplains provide and cope with the additional threats that the floods can pose to property and life. Traditions, ways of living and moving, market dynamics, cultural activities, and internal rules on how to manage the floodplain and shared natural resources are strongly shaped by the annual cycle of flooding. In the Sahel, they form the backbone that links and sustains both wetland and adjacent and nomadic dryland societies [4].

Increasingly, flood plain hydrological regimes are being impacted by increased demand and utilization of upstream river basin ecosystem services. Increasing energy demand in SSA driven by increasing population is underpinning the growth of the hydropower industry. The Food and Agriculture Organization of the United Nations report that there are over 48 dams with a reservoir capacity of more than 1 km$^3$ that affect river connectivity and discharge dynamics [5], especially magnitude, timing, and duration of the flooding regime. While SSA's food security is mostly met by rain-fed and flood recession farming, about 3.5% is met through irrigated agriculture and rising [6]. In some West-Sahelian river basins, water quality is being degraded due to untreated wastewater discharge and irrigation return flows [6]. Overgrazing of the land and deforestation is driving increased sediment yield, leading to widespread siltation of waterbodies and river channels. Exacerbating this, climate change is set to compound these changes in the natural flooding dynamics, a decline in per capita water availability in Mali by 77% by 2080 compared to 2000 [7].

The impacts of these changes on floodplain systems are serious, if not always immediately obvious. Seasonal flood regimes drive ecosystem processes that underpin both natural and local scale managed production systems. Flood recession rice germination depends on flood timing in relation to soil moisture and production on flood extent to reach the land that communities intend to cultivate. Growth of grass species such as Bourgou, used extensively by nomadic and sedentary livestock raisers, depends on a depth of flooding whilst production depends on flood extent. Natural fisheries require sufficient flooding to create conditions for spawning and provide habitats for fish nurseries. When flood regimes change due to upstream allocations or changed timing, reduced flood extent can drive a loss of production. Where changes cross certain thresholds such as the depth of water needed for Bourgou to

grow then changes are non-linear. Regrettably, most contemporary water resource management plans and operations fail to take this knowledge fully into account. There is a need for tools that embed floodplain ecosystem needs into guidance, enabling better informed decisions that are accepted and used. The concept of environmental flows (e-flows) acknowledges the linkages between river dynamics, ecosystem functioning, and the potential of delivering ecosystem services that sustain livelihoods. Environmental flows are defined as the quantity, timing, and quality of water flows required to sustain freshwater and estuarine ecosystems and the human livelihoods and wellbeing that depend on these ecosystems [8]. In recent years, e-flows have been established across the world in various river basin contexts. They are considered a useful policy instrument by water resources managers to manage shared water resources in the food–water–energy nexus and to include environmental sustainability aspects in water allocation choices. Examples from Africa abound [9], such as the Okavango in Botswana [10], the Lesotho Highlands and Mara in Kenya [11,12], and several rivers in Tanzania [11]. To establish effective e-flows that have wide support and compliance, a strong water governance approach rooted in IWRM and especially environmental management is necessary [13,14].

In many SSA countries, water governance is in practice, still being developed [15]. Often, countries lack sufficient technical capacity to manage water resources effectively, and water allocation decisions are not always based on the underling evidence. Implementation of IWRM is hampered, with many government institutions offering mandated but limited meaningful integration in many countries. In addition, there may be a political economy behind the choices that are made that are non-transparent to a wider audience. For example, compliance with water allocation agreements and dam operating rules can be weak. The needs of water users that are inadequately included in decision-making are often ignored.

In this paper, we argue that e-flows should be central to the management of SSA floodplains and their productive systems. To do so effectively, the consideration of often-marginalized primary users of water resources in water governance is essential. Instead of focusing only on the more ecological dimensions of e-flows, it is better to see them as a tool to achieve wider societal goals [16]. In SSA, where water is at such a premium and under heavy pressure from socio-economic, food security, and local scale users, the first question water decision-makers would need to discuss is how do they want the floodplain societies (and economies) to develop: how much food and or GDP should they generate, what kind of livelihood should they be able to sustain in order to create stability? Additionally, what level of flood risk is acceptable and how to manage the impacts of climate change? The next question would be how this relates to healthy functioning ecosystems such as floodplains and, hence, in what state they should be. Such an approach would allow societal goals to be satisfied by a combination of natural systems such as healthy floodplains together with grey infrastructure such as irrigation and dam development. This combination of green and grey infrastructure is now recognized to be essential for sustainable development particularly in Africa [17].

The Inner Niger Delta (IND) located in Mali is one such example where this combination can be embraced. The Niger River, rising in the moist highlands of Guinea, feeds into the IND, the second largest floodplain wetland in Africa after the Sudd Wetland in the Nile River Basin extending some 400 km in length and 100 km in width. Located on the southern edge of the Sahara Desert, this floodplain is home to many farmers, herders, and fishers: a million and a half people fully depend on the exploitation of the natural resources of the IND including rice farmers (5000–170,000 t floating rice/year), cattle breeders (2 million cattle and 5 million small ruminants), and fishers (50–100,000 t/fish/year). The flooding of the Delta is the engine of the socio-economic development as well as its ecological integrity [18]. Relatively small changes in the amount of water entering the delta during the flood period can have a large effect on the size of the extent of flooded floodplains ranging from about 10,000 km$^2$ in drier years up to 20,000 km$^2$ in wet years (Wetlands International, 2020). The IND is not only vital for the local economy, it is also important for the national and regional economy. It is estimated that the IND provides 30% of Mali's rice, 80% of national fish production, as well as dry-season grazing for up to 60% of Mali's cattle. Pastoralists and

cattle from some neighboring countries move into the IND in the dry season, while fish from the IND is exported across West Africa.

Countries such as Mali have a "river-dependent economy" that is driven by seasonal variation in rainfall and river flow. Experiencing often high year to year climate and therefore hydrological variability can also have a significant impact on production. A popular solution to this climate dependency has been the development of dams with associated hydroelectric and hydro-agricultural irrigation schemes [19] (see Figure 1.1). Various policies and plans for the sustainable development of the IND exist; however, their good intentions do not always materialize, as the country does not fully embrace the level of water governance required for operationalization of such plans. Management of water resources, rural development, and ecosystems are distributed over various ministries, with the Ministry of Water and Energy building IWRM capacity and in the process of establishing new water policy. Catchment management plans are also under development but implementation lags.

IWRM coordination takes place through various mechanisms such as the commission gestion des eaux de la retenue de Selingué et de Markala, (CGESM, commission for the management of the reservoirs of Selingué and Markala) and through regular interdepartmental meetings. Mali, as one of the Member States of the Niger Basin Authority, adopted the Niger Basin Water Charter that came into force in 19 July 2010. To guarantee sufficient water resources for drinking water supply to downstream Niamey in Niger, a minimal flow release of 50 m$^3$/s into the IND from the Markala barrage was agreed.

Many human communities living within the high-risk flood-driven environment of the IND continue to live with uncertainty that often threatens their livelihoods. Although many of these communities have survived during wet and dry years, they are still largely dependent on and vulnerable to the natural flow regime for crop production, pastures, fishing, and natural resource harvesting. More recently, these systems have also been put under increasing pressure from a growing population driving degradation of the resources and increasing competition. Since 2012, food insecurity in the delta has been rising mostly resulting from bad governance, conflicts, and access to markets, for example, not necessarily due to reduced supply of ecosystem services. Only during observed severe drought periods, as experienced in 1984, this resulted in severe food insecurity in the delta and regional Sahel.

The determination of e-flows and their implementation in mainstream water resource management planning and operations of the UNR and IND can contribute to a sufficiently healthy functional floodplain ecosystem to provide the ecosystem services required to sustain its flood-dependent society and economy. The research question for the study queries if suitable, holistic e-flows can be established for the UNR and IND on appropriate spatial scales that address social and ecological features and values of the system that will contribute to sustainable floodplain management for the people and environment of the IND.

The aim of this paper is to demonstrate the contribution that holistic regional e-flow assessments can make to floodplain sustainability where the e-flows requirements and consequences of altered flows can be linked to floodplain ecosystem and human livelihood management in the Sahel of North Africa. In this study, the holistic PROBFLO approach has been implemented to determine the e-flow requirements for the Upper Niger River and Inner Niger Delta (UNR and IND) and evaluate the socio-ecological consequences of altered flows associated with a range of water resource use scenarios. PROBFLO is a form of regional scale ecological risk assessment developed to evaluate the probable negative effects of flow alteration and other non-flow stressors, affecting dynamic ecosystems on multiple spatial scales [12,20]. The foundation of the approach follows development by [21] and [22] and includes undertaking regional scale ecological risk assessments using the relative risk model (RRM) and Bayesian network (BN) probability modelling methods [12,20,23] This RRM-BN approach incorporates probabilistic models of socio-ecological systems and the cause and effect risk pathways of multiple sources to stressors to receptors within a range of habitats that ultimately drive ranked socio-ecological impacts or endpoints in a holistic manner [12,20,23].

The relative risk of the flow and non-flow stressors is calculated for social and ecological endpoints selected to represent the ecological and ecosystem service features of the landscape (such as the flood-dependent economy in the IND) that we want to manage [20]. The RRM-BN approach is a transparent, adaptable, and evidence-based probabilistic modelling approach that can also incorporate expert solicitations and explicitly address uncertainty. This approach has successfully been used throughout Africa to evaluate the effects of altered river flows including the quantity and timing of flows (or e-flows) needed to meet the sustainable requirements of freshwater and estuarine ecosystems and non-flow variables [8,12,20].

For the application of PROBFLO for the UNR and IND, available water resource use, ecosystem service, and ecosystem information, with limited ecosystem driver evaluation, have been used to implement the ten procedural steps of PROBFLO [12]. For this assessment, the present-day river flow volume, timing, duration, and frequency characteristics, representing the "present condition" for which relevant present-day bio-physical, ecosystem service and process data can be collected and evaluated, has been evaluated. These "present condition" outcomes were compared to "reference flow" conditions, representing pre-anthropogenic development conditions or early twentieth century conditions and used to establish suitable risk thresholds that provide flow and non-flow requirements to meet these thresholds that represent the e-flow requirements [12]. Four additional alternative water resource development scenarios were included in the assessment to evaluate the socio-ecological consequences of alternative water resource use options. This paper presents the implementation of PROBFLO to establish the e-flow requirements for the UNR and IND and evaluate the risk of altered flows to contribute to the sustainable management of the IND and the vulnerable ecosystems and people who depend on it.

## 2. Methodology

The study area includes the bifurcated Niger River and its main tributary, the Bani River that flows in a North-Easterly direction towards the IND (Figure 1). The ten procedural steps of PROBFLO were implemented to determine the e-flows and risk of various scenarios related to alternative water resources management scenarios in the UNR and IND. Following [12], the procedural steps of the risk assessment include the establishment of a vision (step 1) for the water resources being evaluated, which resulted in the selection of social endpoints associated with the maintenance of the livelihoods of local communities, and ecological endpoints that address biodiversity and ecosystem processes of the resources. Thereafter, a literature review was undertaken for the study area and maps were established of water resources and associated ecosystem services (step 2). The study area was then divided spatially into eight geographical risk regions (Figure 1), allowing the ecosystem dynamics and endpoints to be evaluated in a relative and spatial manner (step 3). In step 4, conceptual models that demonstrate the causal risk pathways from identified sources (including anthropogenic and natural activities/events) to stressors (water quality, flow and habitat modifications, for example), socio-ecological receptors in multiple habitats to endpoints, were developed. A ranking scheme was established to represent the condition of each variable of the study and risk to endpoints (step 5). The risk was calculated (step 6) using Microsoft ® Excel (Microsoft corporation, https://office.microsoft.com/excel), Netica (by Norsys Software) to construct BN and determine the distribution of risk ranks that represent the risk profiles for each endpoint. These outcomes were then combined through multiplication of random assignments of risk ranks, based on endpoint probability distributions obtained from the BN for each of the four ranks used in the study, into meaningful integrated social or ecological risk probability distributions for each risk region using Monte Carlo procedures undertaken with Oracle Crystal Ball software (Oracle, Oregon). These randomization evaluations were also used to quantify the effects of parameter uncertainty on the risk predictions [24], with sensitivity evaluation procedures in Netica for uncertainty testing in this assessment (step 7). A monitoring plan/program was required so that management could test the validity of the risk assessment (step 8), which was then tested by implementation of management and the corresponding monitoring (step 9). The last step of the approach was to

communicate the outputs of the risk assessment and generate good practice recommendations for future sustainable management and risk mitigation (step 10).

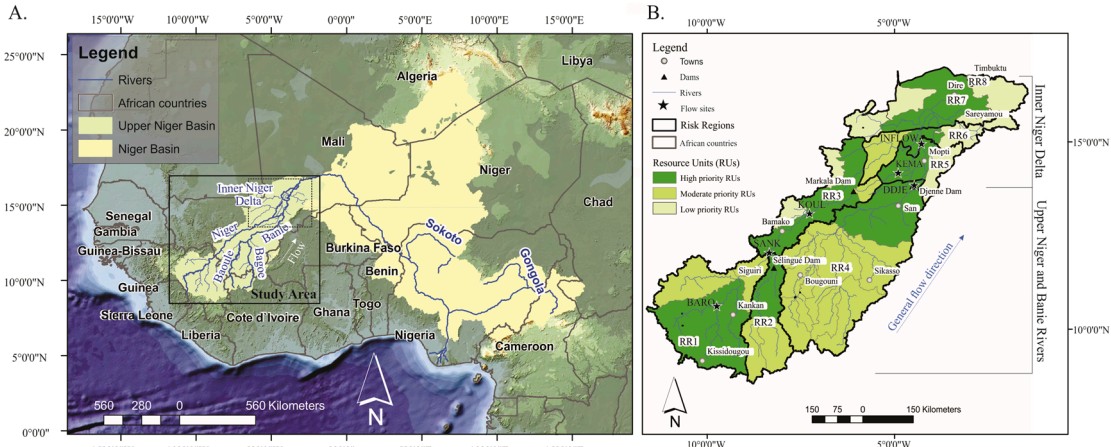

**Figure 1.** The Upper Niger River and Inner Niger Delta, West Africa including the spatial extent of the basin (**A**) with the Inner Niger Delta highlighted. Additional map of the resource units selected to determine the environmental flows and socio-ecological consequences of altered flows in the basin (**B**). The Inner Niger Delta floodplain delineated as risk region 6, 7, and 8.

## 2.1. Vision and Endpoints

Stakeholder views and priorities for the river/floodplain must be translated into a clear, shared vision and formal objectives for environmental water management and, indeed, more broadly for water resource management [25]. They provide the definition "The vision represents the social narrative statement, which may contain specific numerical parameters, of the desired state of the river (resource): What do we want the river (resource) to look like?" [25]. Objectives may then provide greater detail, e.g., that the fisheries should be sustainable. While an extensive stakeholder consultation would be ideal, in this project, it was not possible to consult widely, so the project had to rely on statements of vision as contained in policy and management documents. Such statements have the advantage of being published documents thus have a certain legitimacy but suffer the shortcoming that often they only make oblique reference to the requirements for water flows and associated ecosystem services.

Key statements of vision are provided by the Office Du Niger through the Water Charter [20] and the Master Plan (2005–2020) that include the need for minimum e-flows to allow water to be accessed by both upstream and downstream countries equitably. They detail that a flow rate of 50 m³/s downstream of the Markala Dam is required, based on studies that were conducted after the 1985 drought. They also stipulate water for agricultural purposes, irrigated rice production must be increased to 1 million tons per annum and the area under irrigation to over 900,000 hectares. They plan that sufficient water is available and that flooding patterns that negatively affect farmers are managed. Objectives for important flood crops *Oryza glaberrima*, or "floating rice", and Bourgou grass used for grazing are both included as objectives [18].

They state that naturally grown fish should provide 50–100,000 t/fish/ year [18], while water for navigation was also identified as a key objective. The Ramsar status [26] is also to be maintained based on its importance for water-bird migration and residence.

The Government of Mali in a contract plan with the Office du Niger (2014) was to implement many of these objectives through improvement of water management, regulation of agriculture, and rehabilitation of irrigated areas and infrastructure [27]. They also assumed the responsibility to improve agricultural production and to increase rice production to 1,122,350 tons in 2018 [28].

The Water Charter [28] also sets out to preserve the quantity and quality of water resources and the sustainable use of water, based on the long-term protection of available water resources and the

aquatic environment. This includes prevention of pollution and events such as floods, droughts, siltation, and climate change.

The Mali Government Investment Plan (Niger Basin Authority, 2012) aims to increase the area equipped for irrigation and rehabilitate existing irrigation schemes in order to increase and secure agricultural productivity through rehabilitation and construction of dams and creation, development, or rehabilitation of 169,970 ha of agricultural land, and, in the longer term, an additional 900,000 ha irrigated land. They also plan to manage fish stocks and develop fish-farming, support livestock management including through construction/improvement of infrastructure, and support grazing.

Regulators will prioritize drinking water supply through infrastructure and the treatment of urban and industrial wastewater. They also plan to manage water-related diseases through an awareness raising campaign and to manage forest areas sustainably including a 30% decrease in firewood harvesting.

An objective was to increase the protection of the aquatic environment against degradation (including invasive plant species), through implementation of management plans for four RAMSAR sites. Additionally, in other sites, conservation actions will be designed.

There is a plan to carry out integrated watershed protection actions in order to reduce erosion and to thus silting in reservoirs and in the natural hydrographic system, and in order to improve farm system performance and sustainability. Another objective would be to reduce sources of pollution from polluting activities (mining, oil production, and farming, etc.).

## 2.2. Endpoints

Endpoints have been defined as "specific entities and their attributes that are at risk and that are expressions of a management goal" [29]. In the case of the Upper and IND, these respond to the vision. Thus, for example, if the vision and objectives for the system include that the floodplain continues to provide fish as a protein source for the local inhabitants, then the endpoint will be that fish are indeed being provided by the system to be used by local inhabitants. PROBFLO estimates the risk of failure of this provision.

A number of endpoints are defined for this study, three social and nine ecological (Table 1) that are at the intersection of natural resources and the livelihoods of people that are at risk as a result of changes to the system. Some of the endpoints are purely socio-economic in nature, while others are purely ecological and represent aspects of the ecosystem that need to be maintained to ensure a fully functional ecosystem, which in turn will reflect on the provision of livelihoods to the people.

The ecosystem service requirements of stakeholders in the IND have quantitative requirements related to the area/extent of the floodplain that is available to provide services, which is also linked to the area of habitat available for ecosystem resilience. This can be compared to the qualitative approach to establish e-flows for rivers that tend not to consider the quantity of ecosystems and where reaches of river are selected to represent all of the rivers in a RR [12].

**Table 1.** Social and ecological endpoints that represent the socio-ecological features that stakeholders care about to demonstrate sustainable use and protection of the Upper Niger River (UNR) and Inner Niger Delta (IND).

| ENDPOINT NUMBER (DESCRIPTION) | DESCRIPTION |
|---|---|
| Social Endpoints (SE) | |
| SE1: (Vegetation for society) | Maintain production potential of indigenous vegetation and subsistence agriculture to sustain community livelihoods. |
| SE2: (Subsistence fisheries) | Maintain fisheries production for community livelihoods. |
| SE3: (Water disease) | Manage water disease to ensure that there is no increased risk of waterborne disease to local communities. |
| Ecological Endpoints (EE) | |
| EE1: (Floodplain vegetation) | Maintain habitats and ecosystem processes for indicator floodplain macrophytes (specifically specializes floodplain species for the IND floodplain). |
| EE2: (River riparian vegetation) | Maintain habitats and ecosystem processes for indicator riparian vegetation (specifically base flow and flood indicator species for the Niger and Bani Rivers considered in the study). |
| EE3: (River indicator invertebrates) | Maintain habitats and ecosystem processes for indicator invertebrates (focused on specialist rheophilic spp. for riverine section of study). |
| EE4: (Floodplain indicator invertebrates) | Maintain habitats and ecosystem processes for indicator invertebrates (including wetland indicator species and limnophilics for the IND floodplain). |
| EE5: (River indicator fish) | Maintain habitats and ecosystem processes for indicator fishes (focused on specialist rheophilic spp. for riverine section of study). |
| EE6: (Floodplain indicator fish) | Maintain habitats and ecosystem processes for critical indicator fishes from the IND floodplain (including wetland indicator species and limnophilics for the IND floodplain). |
| EE7: (Aquatic mammal indicators) | Maintain habitats and ecosystem processes for indicator aquatic mammals (Manatee *Trichechus sp.* populations (specifically for the IND floodplain). |
| EE8: (Resident indicator birds) | Maintain habitats and ecosystem processes for indicator resident birds from the Niger and Bani Rivers and IND floodplain. |
| EE9: (Migratory birds) | Maintain habitats and ecosystem processes for indicator migratory birds from the IND floodplain. |

*2.3. Sources and Stressors*

There are several anthropogenic activities (sources of stressors) that are affecting a change in the flow and quality of the UNR and IND. These include existing dams (such as Markala, Sotuba, Selingue, Talo, and Djenne) and possible new ones such as the one planned near Fomi in Guinea for water supply to irrigation and hydropower generation that changes the flow patterns downstream of these dams [30]. Various irrigation schemes are being operated of which the Office du Niger upstream of the IND at Markala is the largest one with a current area of about 1200 km$^2$ and a plan to expend to 4600 km$^2$ in 2045 [31]. The spread of human settlements has resulted in the formation of stressors including over-grazing and general agricultural development with resultant land degradation. Other sources of stressors in the basin include mines, small and large urban developments such as Bamako, and an increasing population of people [17], all of whom have their own requirements for the resources of the basin. In the PROBFLO study, a conceptual model that includes components of the socio-ecological

systems and relationships between components was developed (Figure 2) and used to facilitate the development of the Bayesian network (BN) probabilistic models.

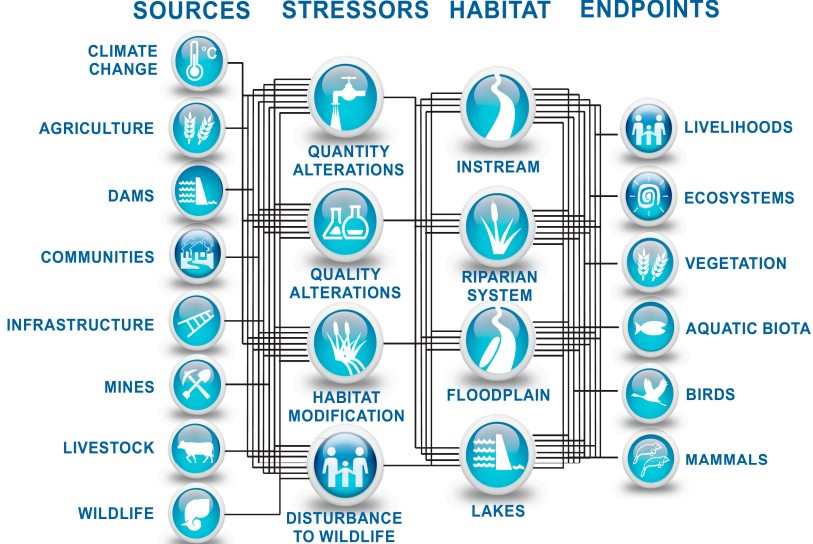

**Figure 2.** Conceptual model schematically representing risk pathways between sources, stressors, habitats (that contain multiple receptors), and endpoints for the study.

*2.4. Risk Regions*

For the selection of risk regions in this study a combination of the management objectives, source information, water resource developments, and available habitat data were used to delineate spatial geographical risk regions for the relative risk assessment [12,24]. This allows the outputs of the assessment to be presented at a spatial scale with multiple regions compared in a relative manner. Through this approach, the dynamism of different regions can be incorporated into the study and allow for a holistic assessment of flow and non-flow variables. The approach can address spatial and temporal relationships of variables between risk regions, such as the downstream effect of a source of stress on multiple risk regions, in the context of the assimilative capacity of the ecosystem or the requirements of ecosystem response components, e.g., fish. Risk results per region for a range of alternative water resource use and protection scenarios also allow stakeholders to consider trade-offs between the socio-economic value of development and the socio-ecological costs in the form of risk to endpoints that represent what stakeholders care about in the landscape [20].

The risk regions (RRs) selection for this study were delineated using a number of criteria, including hydrological catchment boundary considerations. An additional hydrological consideration was to select regions where changes in flows from natural/reference to present day or future due to developments (dam construction, irrigation, or hydropower) could be assessed by the socio-ecological scientists in the study. The final number of RRs selected was eight with five in the upper catchments of the Niger and Bani Rivers, two within the IND, and one at the outlet of the IND at Timbuktu. The details of the selected RRs and hydrological rationale for the selection is provided in Table 1.

To support this process the RRs have been sub-divided along catchment, ecological habitat, and social dependence activities into discrete, manageable spatial areas or units [12]. These discrete units are generally socio-ecologically homogenous in nature and have been referred to as resource units (RUs). In this study, 36 RUs were delineated using available basin boundaries, ecoregions, river/stream classification (geomorphological classification), habitat and associated ecological and ecosystem service scenarios, water quality trends, and water resource use scenarios (Figure 1). Due to the high number of RUs and data limitations to describe socio-ecological features and processes for each RU, a rationalization process to prioritize and select the most useful RUs that represent what stakeholders care about in the basin was incorporated in the study. This was achieved using a decision

support tool and stakeholder engagement (supplementary information). In this study at least one RU has been selected to represent each RR (Figure 1).

### 2.5. Conceptual Model

The conceptual model developed in step 4 is a critical step that describes the cause–effect linkages for all the evaluated risk components including; the sources, stressors, habitats, and impacts to endpoints selected for the case study (Figure 2) [12,21,32,33]. The model includes the holistic (considering flow and non-flow related variables, e.g., water quality, in a spatial–temporal context), best practice characterization of flow-ecosystem and flow-ecosystem service relationships in the context of a regional scale e-flow framework [12,34]. Conceptual models were constructed during an expert workshop after the completion of the literature review of the UNR and IND. The workshop included hydrologists, geomorphologists, ecologists, and ecosystem services scientists. They were able to generate hypotheses that represent the socio-ecological processes of the system being evaluated and probable cause and effect relationships of: (1) sources to stressors to (2) multiple receptors in relation to (3) their impacts on the endpoints selected for the study. The conceptual models addressed the requirements of the PROBFLO approach. The PROBFLO conceptual model thus conforms to the regional scale e-flow framework procedures in: (1) the selection of socio-ecological endpoints, to direct the hydrologic foundations for the study including the selection of hydrological statistics required, (2) to classify ecosystem types based on geomorphic, water quality, quantity, and ecoregion considerations, and with these data, (3) to incorporate evidence-based flow-ecosystem relationships and flow-ecosystem service relationships, with relevant non-flow variable relationships upon which the assessment is based.

Following the conceptual model development workshop, a master conceptual model highlighting major risk pathways was initially developed (Figure 2). Thereafter detailed models were established for the endpoints selected in the study (example Figure 3). These conceptual models represent our understanding of the relationships between sources and endpoints in the study and can be adapted with new information. The detailed conceptual models were used to generate BN models for each endpoint that were integrated into a master BN (Figure 4).

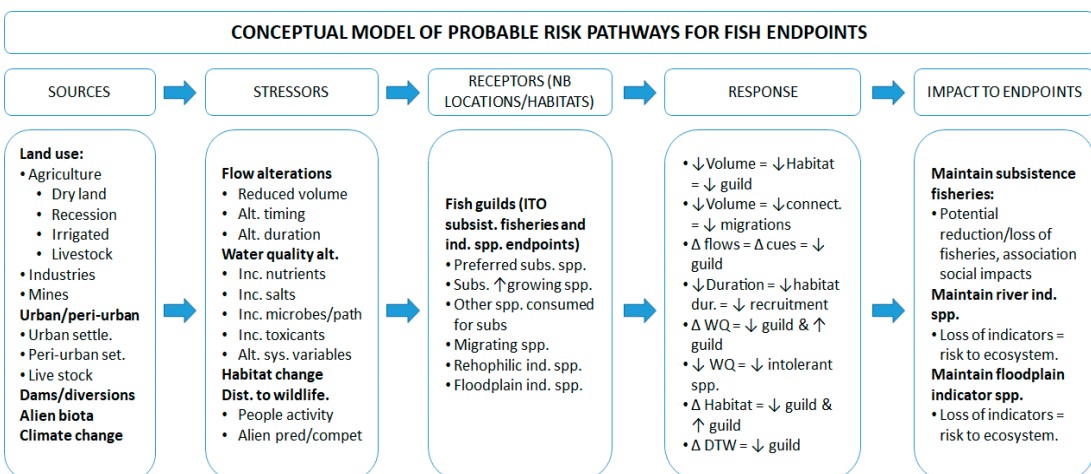

**Figure 3.** Detailed conceptual model for fish used to direct the formation of the risk model for fish endpoints.

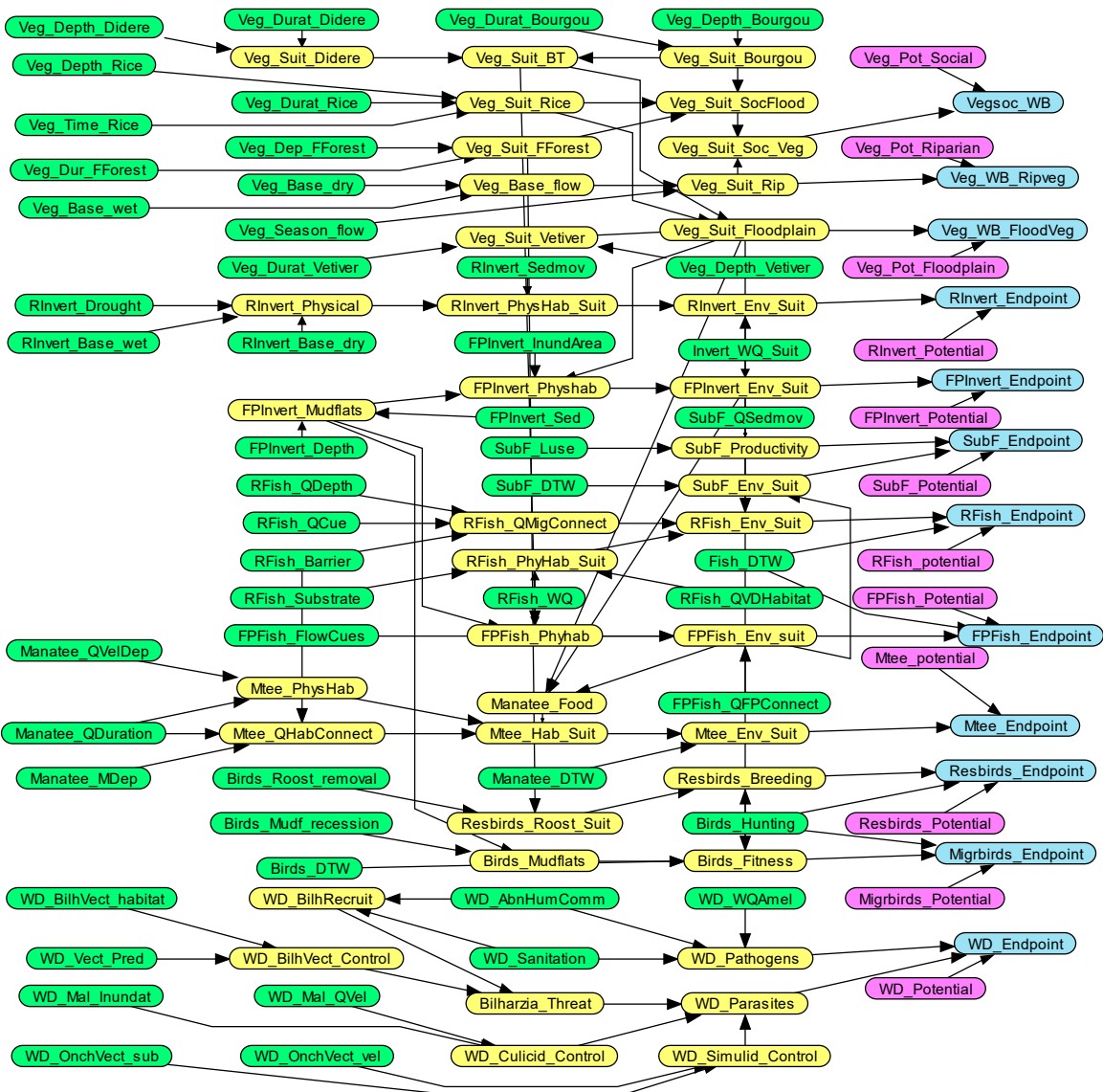

**Figure 4.** Integrated Bayesian network used as the risk model for the study to represent the risk pathways and socio-ecological response of endpoints in the study to altered flows and non-flow determinants using Netica Bayesian Network software by Norsys Software.

### 2.6. Ranking Schemes

Ranking schemes (step 5) facilitate the calculation of relative risks to each social and ecological Endpoint (supplementary information (SI) Table S1) [32]. The four states that are commonly used in RRMs, namely zero, low, moderate, and high [20,23,32,35,36] have also been incorporated into the PROBFLO process. The states represent the range of conditions, levels of impact, and management ideals [12] as follows:

- Zero: pristine state, no impact/risk, comparable to pre-anthropogenic source establishment, baseline, or reference state;
- Low: largely natural state/low impact/risk, ideal range for sustainable ecosystem use;
- Moderate: moderate use or modified state, moderate impact/risk representing threshold of potential concern or alert range;
- High: significantly altered or impaired state, unacceptably high impact/risk.

In this risk assessment, ranks including zero, low, moderate, and high were assigned threshold relative risk scores of 25, 50, 75, and 100 for the BN and Monte Carlo randomization evaluations to integrate the social and ecological components of the study. This ranking scheme represents the full range of potential risk to the ecosystem and ecosystem services with management options [12]. By incorporating BN modelling into RRM-BN, the variability between ranks for each model variable can be represented as a percentage for each rank and are assigned scores along a percentage continuum representing the state of the variables using natural breaks for rank thresholds of 0.25 (zero), 0.5 (low), 0.75 (moderate), and 1 (high) in the calculation (Table 2) [12].

Data used to parameterize the models for the risk assessment including rank thresholds established to represent the socio-ecological system being evaluated in the study are available in the supplementary information (Table S1). These data include all of the socio-ecological system variables or nodes (node names) selected for the models, network variable titles, ranks, and associated modelling scores, rank definition, and measures for variables and justification for the use of the variables and evidence to describe their use in the risk assessment with references for the evidence used. The quantitative approach to evaluate e-flows for the IND, which is unique to this study, has been incorporated into the RRM-BN where wetland area linked to flow using hydrodynamic modelling [37] has been used to provide measures for the ranking scheme in the BN nodes. The UNR risk region requirements were considered in a qualitative manner where services and associated habitats for selected reaches of river per RR were selected to represent the requirements of the whole RR following the approach for rivers established by [12] and [20].

**Table 2.** Alternative water resource use and protection scenarios selected for the study to evaluate the socio-ecological consequences of altered flows in the Upper Niger River and Inner Niger Delta including descriptions and developments associated with scenarios.

| Scenario | Description (ABR) | Developments Included | Comments |
|---|---|---|---|
| 1 | Reference flows (REF) | Markala weir (1947), Sotuba Dam (1929), and limited irrigation | This scenario represents is the natural flow situation as would have occurred before major developments started to take place. Both Sotuba and Markala account for a very small (~3–7%) flow reduction in the Niger runoff, so are considered insignificant. |
| 2 | Historical flows (HIS) | Markala weir (1947), Sotuba Dam (1929), Selingue Dam (1982), hydropower and irrigation as in 2005. | The present-day scenario (PRS) but excluding the Talo and Djenne Dams on Bani River. |
| 3 | Present day flows (PRS1) | Markala weir (1947), Sotuba Dam (1929), Selingue Dam (1982), Talo Dam (2007) and Djenne Dam (2015), hydropower and irrigation as in 2005. | All present-day developments super-imposed on the reference flows with 50 $m^3$/s release rule downstream of Markala. |
| 4 | Present day flows (PRS2) | Markala weir (1947), Sotuba Dam (1929), Selingue Dam (1982), Talo Dam (2007) and Djenne Dam (2015), hydropower and irrigation as in 2005. | All present-day developments super-imposed on the reference flows with 0 $m^3$/s release rule downstream of Markala. |
| 5 | Future flows 1 (FUT1) | Markala weir (1947), Sotuba Dam (1929), Selingue Dam (1982), Talo Dam (2007) and Djenne Dam (2015), Selingue hydropower and irrigation as in 2045 and Bani irrigation as in 2025. | All present-day and future developments super-imposed on the reference flows. |
| | | Proposed Fomi/Moussako Dam in RR1 and associated irrigation and hydropower. | |
| 6 | Future scenario 2 (FUT2): | Inner Niger Delta specific scenario based on the modelled range of 2018 e-flow minimum (EFA), 70th percentile and e-flow average (EFA) (Dickens et al., 2018b). Flows between the three points were normalised using a linear regression. | This scenario represents a hypothetical high use of the water resources of the IND. The range of flows tested here range from the minimum flows to the e-flow average or 50 percentiles. This scenario represents the provision of critical low flows and base e-flows, without any freshet and flood flows. |
| 7 | Future scenario 3 (FUT3): | Inner Niger Delta specific scenario based on modelled flows ranging from future min (FUT1), E-flows flows min (EFA) and 1984 average flows [38]. Flows between the three points were normalised using a linear regression. | This scenario represents a hypothetical extremely high future use of the water resources of the IND. The range of flows tested here range from the highest hypothetical flow demand obtained from scenario FUT1 to the average e-flows and 1984 observed average base (50 percentile) flows, without any floods. |
| 8 | Future scenario 4 (FUT4): | Inner Niger Delta specific scenario based on maintained minimum average monthly flows observed in the IND during the 1984 drought. | This scenario documents the risks associated with the worst drought in memory if sustained over a longer time. |
| 9 | Future scenario 5 (FUT5): | Inner Niger Delta specific scenario including all dams operated to max hydropower potential, irrigation demand for 2045, and no forced released from Markala. | Future realistic planned development scenario. |

## 2.7. Bayesian Networks

The BN probability modelling in step 6 is used to calculate the relative risk of multiple stressors including altered flows in particular, establish sustainability thresholds for the endpoints, and then, determine e-flow requirements to provide the flows to meet these requirements [12]. The conceptual models were used to establish the BN models applied per RR. Each model includes nodes or indicators of the socio-ecological system and measures to describe the state of each variable [12]. The interactions or relationships between the variables are set up using conditional probabilities and then parameterized, tested, and then applied in the BNs (Table S1). These models were analyzed individually or integrated using a range of BN modelling tools by using nodes representing variables that share the same indicators and measures. The graphic BN models make use of conditional probability distributions to graphically represent the relationships between the variables in the model (Figure 4). The model consists of parent or input nodes that provide the input parameters and child or conditional nodes that receive inputs from one or more parent nodes [20,39]. The interactions between the parent nodes that result in the child node and the probability of all potential outputs based on different combinations of input variables are described in conditional probability tables within the BN [40]. The BN established for this study has been provided as supplementary information (Table S1). Data used in the risk assessment were generated through extensive specialist reviews of available information describing the socio-ecological systems of the UNR and IND, and solicitations from regional Sahelian and Sub-Saharan Africa information presented in International Water Management Institute [38], and reviewed and referenced in the supplementary information.

For the UNR and IND risk calculation steps, the BNs were used to determine current or present scenarios based on available data, field surveys, and expert opinion and then used to model future use and protection scenarios. Scenarios were established by specialists in relation to establishment of e-flows for the study and the consequences of altered flows to the socio-ecological system (Table 3. Some of the scenarios were only relevant to the use of the IND (Table 3). The model was set up using known historical socio-ecological ecosystem wellbeing characteristics compared with current or present-day conditions and then used to model alternative and future resource development and protection scenarios. These scenarios have allowed for the determination of the e-flows for the UNR and IND and consideration of the socio-ecological consequences of alternative water resource use options.



**Table 3.** Risk region selected for the study to evaluate the socio-ecological consequences of altered flows in the UNR and IND including gauging facilities and associated rivers selected for the hydrological evaluations for each risk region (refer to Figure 1).

| Risk Region | Gauging Facilities | Abbr. | River/s | Rationale |
|---|---|---|---|---|
| RR1 | Kouroussa<br>Baro | KOUR<br>BARO | Niger, Tinkisso, Niandan, Milo | Impacts of proposed Fomi Dam in upper reaches of Niger in Guinea important to the study. Flows at KOUR used in the assessment. |
| RR2 | Sankarani | SANK | Sankarani | Impacts of existing Selingue Dam important. Flows at SANK used in assessment |
| RR3 | Koulikoro | KOUL | Niger | Both Sotuba and Markala weirs in this RR. Flows at KOUL downstream of Sotuba weir used for assessment. |
| RR4 | Bougouni<br>Pankourou<br>Dioila<br>Koroudougou<br>Douna<br>Talo | BOUG<br>PANK<br>DIOL<br>KORO<br>DOUN<br>TALO | Baoule<br>Banifing<br>Bani | Entire Bani catchment to just upstream of Djenne Dam. Flows at TALO used for assessment. |
| RR5 | Ke Macina<br>Sofara<br><br>After confluence | KEMA<br>SOFA<br><br>INFLOW | Niger<br>Bani<br><br>IND | Impacts of Markala weir on lower Niger.<br>Impacts of Talo and Djenne Dams on lower Bani.<br>Flows at KEMA (Niger), SOFA (Bani), and after confluence (INFLOW) used in assessment. |
| RR6 | Inner Niger Delta | | | Part of IND–hydro-dynamic model. |
| RR7 | Inner Niger Delta | | | Part of IND–hydro-dynamic model. |
| RR8 | Outlet of IND | DIRE | Niger | Observed flows at Dire (1979–2007) used in assessment. |

### 2.8. Uncertainty

The RRM-BN approach includes an evaluation of uncertainty (step 7), so as to identify key drivers in the model and sources of uncertainty that may be impacting the overall uncertainty of the model [20,22]. The results of the uncertainty evaluation provide context for the stakeholders and contribute to the water resource management decision-making process. The successful establishment and testing of risk hypotheses allowed the RRM to be validated, which reduced overall uncertainty. This included application of the "Sensitivity to Findings" tool of Netica to evaluate the contribution of individual variables (nodes) to the risk outcomes and the Monte Carlo randomization approach in Oracle Crystal Ball software to integrate and test random effects of risk predictions [12].

In addition, various contributory methods including the use of geographical information systems to facilitate mapping, exposure, and effect pathway establishments, as well as the use of Monte Carlo and Bayesian techniques to address uncertainty have been developed to complement, validate, and strengthen this approach [21].

## 3. Results

### 3.1. Descriptive Results from Existing Flow Variability

The primary result of reducing flows upstream of the IND includes a reduction in the magnitude and extent of flooding in the IND. This would lead to a reduced wetted area and, hence, a reduced quantity of all of the components of the ecosystem that depend on a wetland habitat. The historical record shows flooding variability (inundation extent) between wet, normal, and dry hydrological cycles that last numerous years, showing that the natural ecosystem has the capacity to endure these extremes events (consider 1984 extreme drought). However, currently, nature is only called to do this on infrequent occasions and without multiple additional anthropogenic stressors. The additional pressure imposed by upstream developments removing water from the system and potential climate variability resulting in greater frequency and persistence of droughts mean that the IND floodplain will be subject to these stress cycles more often without any assurance that it will be able to recover sufficiently between stress cycles. At what point will the benefits from the floodplain become so compromised that it loses its value in the region? These stress cycles will impact on the IND ecosystem and will affect all the beneficiaries of ecosystem services in the region. Changes to the overall extent of the ecosystems can be illustrated graphically by documenting the inundation area that has already been experienced during different historical flow periods (i.e., very dry in 1984; moderate in 1995; wet in 1994) together with the future scenarios for development (Figure 5). Not only does the inundation (depth) of the floodplain change considerably, but also the speed of water flow (velocity) that would have a knock-on effect on other components of the ecosystem. Moving water in turn moves the sediments, flushes rotting vegetation, prevents invasion by reeds, and provides many habitats for biota.

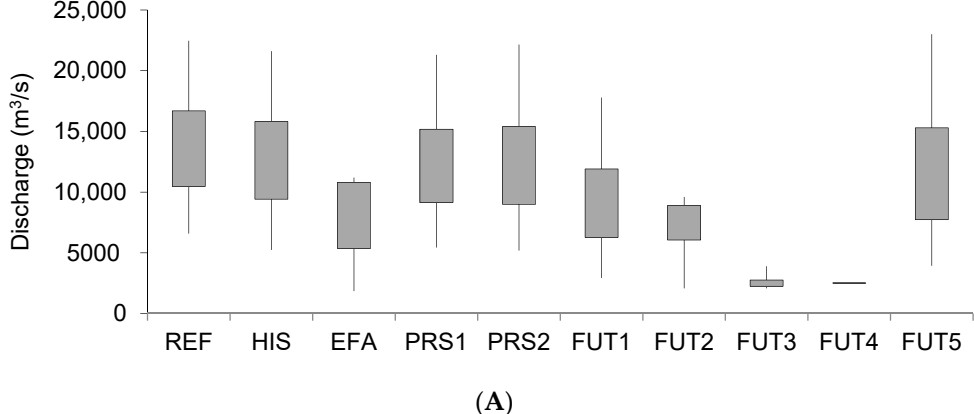

(**A**)

**Figure 5.** *Cont.*

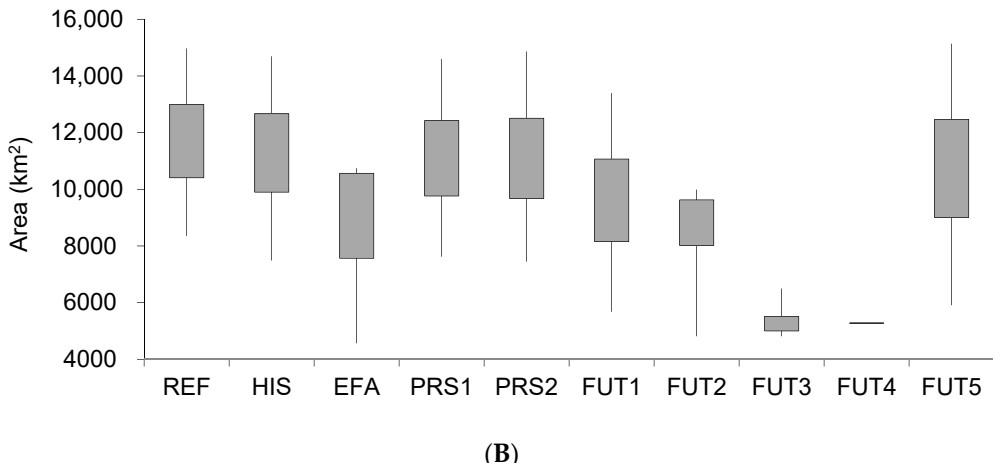

**(B)**

**Figure 5.** Box and whisker plots of the total flow ((**A**), discharge in m³/s) variability of each of the ten flow scenarios considered in the study for the evaluation of the socio-ecological consequences of altered flows in the UNR and IND. The boxes represent 20–80 percentiles and whiskers represent range from 0.01 to 99.9 percentiles and for inundation area the (**B**) in km² of inundation variability in the IND for each of the ten scenarios considered in the study.

### 3.2. Environmental Flows Determined

In the context of e-flow requirements, seasonal flow variability over the course of the year is essential for the maintenance of the ecosystem [41]. The volume, timing, duration, and frequency of seasonal flows in the UNR and IND is associated with essential requirements of the ecosystem response indicators, the vegetation, fish, and invertebrates as well as the social needs associated with the ecosystem. The e-flow determination process in PROBFLO used the ranking schemes and thresholds of sustainability established in the RRM-BN assessment to determine integrated e-flow requirements for the UNR and IND (Table 4). The e-flow requirements have been included in the socio-ecological consequence assessment of altered flows to the IND in addition to development or protection scenarios.

**Table 4.** Summary of the total environmental flow requirements and flows required to maintain base (low) flows, droughts, and floods for the five risk regions in the UNR and IND.

| Risk Region | River | Reference (MCM) | % Requirement | | | | Volume Requirement (MCM) | | | |
|---|---|---|---|---|---|---|---|---|---|---|
| | | | Low Flows | Drought Flows | Floods | Total | Low Flows | Drought Flows | Floods | Total |
| RR1 BARO | Niandan | 6759 | 28.6 | 9.0 | 10.7 | 39.3 | 1950 | 611 | 727 | 2678 |
| RR2 SANK | Sankarani | 7641 | 25.6 | 10.5 | 8.9 | 34.4 | 1969 | 811 | 682 | 2651 |
| RR3 KOUL | Niger | 35,458 | 43.5 | 10.8 | 14.7 | 58.1 | 15,550 | 3865 | 5246 | 20,796 |
| RR4 DDJE | Bani | 9148 | 22.8 | 4.7 | 18.1 | 40.9 | 2115 | 430 | 1676 | 3791 |
| RR5 KEMA | Niger | 35,789 | 45.4 | 11.2 | 18.0 | 63.4 | 16,410 | 4058 | 6485 | 22,895 |
| RR5 (INFLOW) * | IND | 45,404 | 40.4 | 9.8 | 17.8 | 58.2 | 18,524 | 4488 | 8162 | 26,687 |

Note: (*) refers to the environmental flow to meet ecological sustainability requirements in the Inner Niger Delta.

### 3.3. General Effects of Altered Flows Associated with Scenarios

To evaluate further the impact of different scenarios on the inundation extent of the wetland, the discharges shown in Figure 6 were converted into inundation areas using the relationships between observed upstream discharges and modelled inundation area using the hydrodynamic model generated in the study [33]. The inundation areas of the IND associated with the different flow scenarios are graphically presented in Figure 6 Inundation areas ranged from 4500 to 15,100 km². The variability range was based on the relationship between inundation area and discharge, so the inundation ranges match the relative differences between scenarios as displayed in the flow variability graph (Figure 6). This low confidence relationship does not account for annual retention capacity of the IND and is only based on observed inundation areas and average monthly flows associated with that area of inundation. An aerial representation of the 3920 km² inundation area observed during the extreme drought of December 1984 (Figure 7A) and the 14,750 km² area observed in the wet season of December 2009 are provided in Figure 7B.

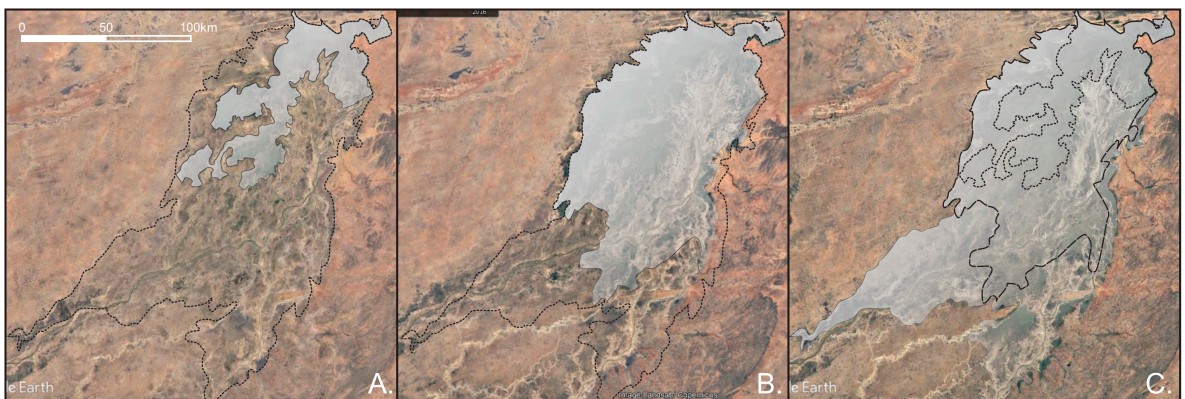

**Figure 6.** Maps presenting the inundation areas (grey areas) of the IND during December 1984 (**A**), 3920 km²), December 2002 (**B**). est. polygon area 11,138 km²), and December 2009 (**C**), 14,750 km²). Generated in Google Earth.

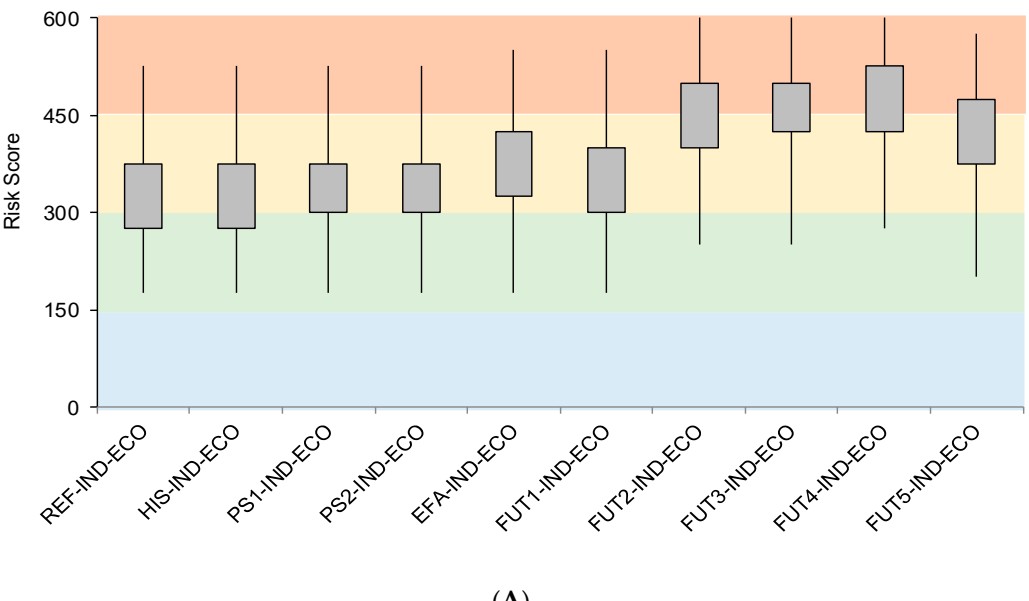

**(A)**

**Figure 7.** *Cont.*

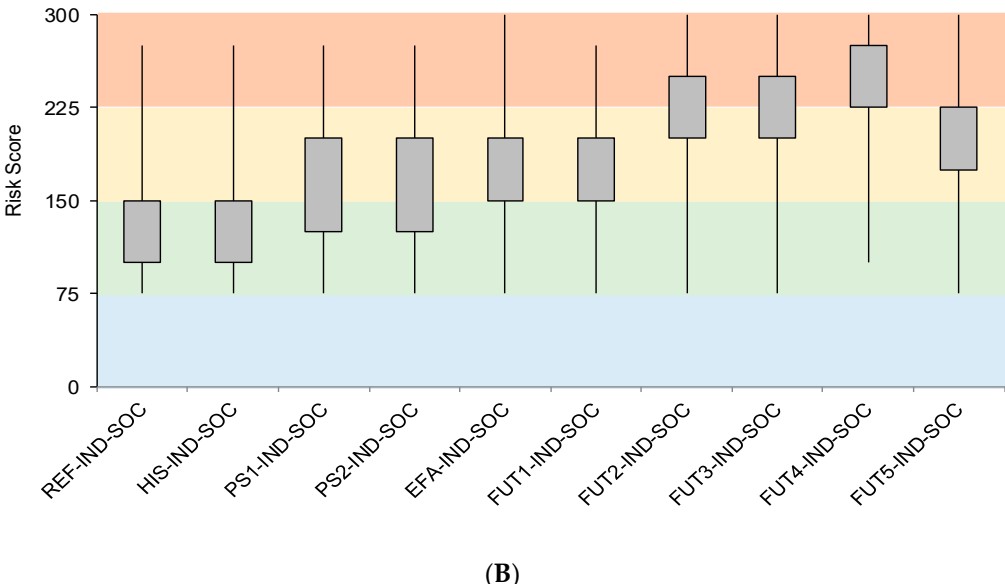

(**B**)

**Figure 7.** The integrated relative risk variability to all of the ecological (**A**) and social (**B**) endpoints considered in the study for each flow scenario considered. Graphs show relative risk score ranges overlaid on zero (blue), low (green), moderate (yellow), and high (orange) risk ranks. The boxes represent 20–80 percentiles and whiskers represent range from 0.01 to 99.9 percentiles.

Due to reductions in water velocity, as well as due to upstream catchment deterioration with associated soil erosion, there is likely to be a change to the movement of sediment through the IND, potentially resulting in the erosion of some river channels, while at the same time leading to the sedimentation of other parts. Changes in the distribution and size of sediment particles would impact on the morphology or shape of the waterways and sandbanks, leading to altered substrate habitats available for other components of the ecosystem and in turn affecting the delivery of ecosystem services. While the IND is naturally variable, these anthropogenic changes will introduce different drivers of ecosystem response. Changes to the hydrology, hydraulic habitat and substrates as affected by the movement of water, as well as water quality, will affect the so-called response indicators, i.e., those aspects of the ecosystem that respond to the drivers of change including vegetation, fish, invertebrates, birds, and mammals.

*3.4. Specific Effects of Altered Flows Associated with Scenarios*

The scenarios considered for this PROBFLO assessment include flow duration tables of monthly averaged data discharge (m³/s) data for the Niger and Bani Rivers as the rivers enter the IND and within the IND (Supplementary information Figures S1–S10). The flow data for each scenario are represented as percentiles in flow duration or exceedance tables (Tables S2–S11). Additional mean monthly average flows (50 percentiles) have been graphically presented with the range of variability associated with the flows represented as box and whisker plots in the supplementary information (Figures S1–S10). In this study, existing alterations in total inflow associated with present (PRS) and future flows (FUT1) into the IND is relatively modest as can be seen in Figure 8 and see Table 2. Planned future developments (FUT1) will reduce the inflow of water in the peak wet season, by which the flood is generated in the IND, by some 20% (Figure 6), and will reduce the dry season flow substantially (supplementary data Figures S1–S10). This project set out to establish if this flow will continue to support the ecosystem and hence the users of that ecosystem as defined by the endpoints (Table 2). In addition, this investigation included four future high development scenarios (FUT2–5) described in Table 3. For FUT2–4 scenarios, the volume and duration of flows were reduced considerably, with FUT4 being the worst-case scenario of a prolonged multi-year 1984 drought. The FUT5 scenario of possible high-level development of

dams and irrigation shows high flood releases but periods of extended, extreme low-to-zero flows in the dry season.

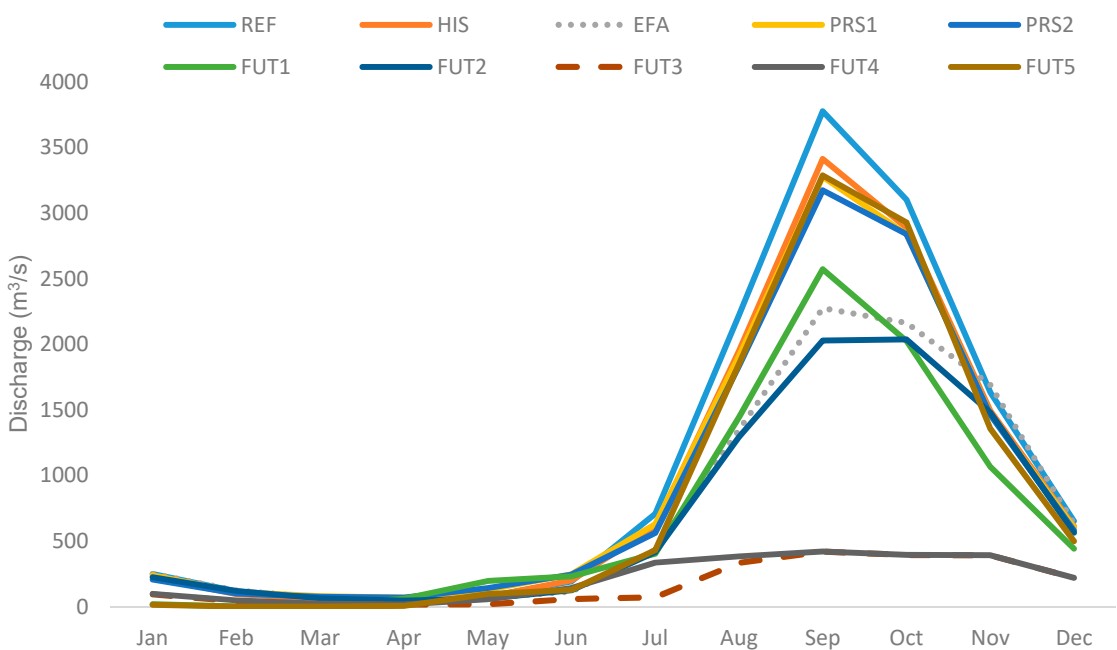

**Figure 8.** Graph of the median (50th percentiles) of monthly average flows (m$^3$/s) for all of the scenarios considered in the study for the inflow to the IND. Graphs represent the different scenarios for reference flows (REF), historical flows (HIS), present scenario 1 (PRS1), present scenario 2 (PRS2), future flows 1–5 (FUT1–5), and (EFA).

Flow variability for each of the scenarios (Table 2) is illustrated in Figure 6; the flows include total annual inflowing flows (as box and whisker plots) from the two main inflowing rivers the Niger and Bani that would enter the IND. The midpoint of each box in Figure 6 represents the mean annual flow. The results include a comparable flow variability between the reference, historical, present scenario 1 and present scenario 2, scenarios that range between 5200 and 22,000 m$^3$/s. Compared to these scenarios the e-flow (EFA) scenario is considerably lower, ranging from 1800 to 11,200 m$^3$/s, although it should be noted that that this still represents 58.16% of the total volume of the reference flows. The future flow scenarios include flow variability (99.9 to 0.01 percentiles) from 2927 to 17,800 m$^3$/s (FUT1), 2100 to 9600 m$^3$/s (FUT2), 2100 to 3900 m$^3$/s (FUT3), and 2500 m$^3$/s (FUT 4). While FUT5 scenario is slightly lower than the reference to present scenarios in terms of variability (3900 to 23,000 m$^3$/s), the FUT1 scenario is considerably better than the e-flow scenario. Future flow scenarios FUT2 to FUT4 are noticeably lower and represent hypothetical flows that can contribute to the evaluation of the resilience and or vulnerability of the IND to severely reduced flows compared to either reference or present flows.

Altered flows in the basin would generally have different impacts on the riverine portion of the basin compared to the IND. Generally, the rivers of the basin will be relatively less vulnerable to reduced flow associated with the scenarios considered in the study. This greater resilience of the rivers can be attributed to the greater extent and diversity of habitat types in the rivers of the basin that will persist with the majority of the flow alterations hypothesised with alternative scenarios considered in the study. One extremely vulnerable attribute of the river section of the study identified in this study is the potential for river connectivity reduction that would affect fish migrations in particular. The outputs of the qualitative assessment of the effect of altered flows on the rivers of the upper Niger and Bani basins include low-to-moderate changes for fish, mammals, invertebrates, and birds, while in the IND, relatively greater impacts have been identified. In the IND, reduced flows have been

directly correlated to reduced inundation extent and habitat loss in the delta. These changes have been evaluated through the qualitative assessment of the IND to result in moderate-to-high risk or threat of impacts to all of the ecosystem components considered in the study, especially for scenarios FUT2, FUT3, and FUT4. The FUT5 scenario will also result in impacts to ecosystem components due to insufficient flows during low-flow periods in particular.

### 3.5. General Risk to the Endpoints in the Inner Niger Delta

The consequences of altered flows associated with high development scenarios include unacceptable, high-risk predictions if scenarios FUT2 to FUT5 were implemented in the study area. This can largely be attributed to the consideration of both qualitative and quantitative measures (specific to the IND section of the study) of the endpoints considered. The results include considerable risk, ranging from moderate (reference flows) to high for e-flows, PRS1, and all FUT flows for the most vulnerable endpoint considered in the study, the Manatee endpoint. Manatees have a high preference for the IND and are particularly vulnerable to habitat change and thus contribute to the risk associated with high development scenarios. Risk estimates show a 40% (±1%) probability that the Manatee or mammal endpoint maintenance is unattainable for all scenarios considered in the study, which demonstrates that the sustainability of these mammals is highly threatened. This is validated by the "vulnerable" status of the species (*Trichechus senegalensis*) in the International Union for the Conservation of Nature (IUCN) Red List. Without conservation efforts, and possibly the establishment of a protected area in the IND, this species is expected to decline and possibly become locally extinct due to the synergistic effects of altered flows, habitat loss, and human disturbance in the IND.

The risk to the ecological (vegetation, fish, invertebrates, mammals, and birds) and social (subsistence livelihoods and water disease) attributes of the system if the timing, volume, and duration of flows were altered have been evaluated per ecosystem attribute. Examination of the risks to the floodplain vegetation show that the vegetation is likely to be tolerant of the small flow changes that are likely to occur within the IND as a result of the development scenarios. This is primarily based on our understanding of the responses of the vegetation and associated habitat characteristics of the IND that have naturally occurred during drought and flood periods. High risk to the sustainability of the vegetation for the IND is proposed for scenarios FUT2, FUT3, FUT4, and FUT5 in particular. These results suggest that the development plans included in the future flows scenario are ecologically acceptable from a vegetation point of view, although it must be remembered that, even though this may be ecologically acceptable, the quantity or area in km$^2$ of ecosystem services provided by the vegetation would reduce, thus the risks to users of the vegetation would increase as the services associated with these endpoints decrease.

The fish communities in the Upper Niger and Bani River sections of the study area will probably have a lower risk of change compared to the fish in the IND. The study does, however, demonstrate the importance of connectivity between the IND and the rivers upstream of the delta and how access for migratory fishes is important to maintain the fish and associated subsistence fisheries in the IND. The results include relatively less risk to the riverine fish communities compared to the IND, where excessive reductions in habitat availability will affect present abundances of fish in the area. Although the risk to the wellbeing of the fish communities and associated human livelihoods in the riverine part of the study area is limited for the scenarios tested (HIS, E-flow, PRS1, PRS2, and FUT1), the risk of losing species and associated processes in the IND is high for scenarios FUT2, FUT3, and FUT4 in particular. For the FUT 5 scenario, many attributes of the fishes of the IND including habitat required to maintain species during low flows would be threatened. All of these scenarios (FUT2, FUT3, FUT4, and FUT5) will result in considerable reductions in fish abundances that will affect human livelihoods in the region. The UNR and IND provides refuge for many fishes that would become threatened if extensive development of the UNR and IND were permitted. This will have consequences for biodiversity and resource protection in the region. Many traits of fishes that have specialist requirements for flowing habitats and those intolerant to altered water quality and flows

would be particularly vulnerable to resource development. Fishes that are vulnerable to fisheries harvesting would also be vulnerable to excessive change to the flows in the IND.

The investigation also considered the risks associated with reduced flows to the invertebrates in the system (river indicator invertebrates and floodplain indicator invertebrates) (Figure S7). The results show a decreasing risk from low to zero in the riverine section of the study area upstream of the IND; while within the IND, there will be an increase in risk to a low–moderate risk (FUT1), and possible collapse of the invertebrate communities for FUT3 and FUT4 in the IND in particular. The invertebrates in the riverine portion of the study are resilient to the changes proposed for the study area, while in the IND area, only major changes associated with scenarios FUT2–FUT4 will affect the habitat availability that the invertebrate communities depend on. Due to reduced habitat in the IND, the abundances and associated consequences of reduced food for fish and birds in the IND are expected.

The IND is well known for its bird populations. The flooded forests and groves along the rivers and within the floodplain play an important role as safe breeding habitat for many resident water birds especially during the flood season. Today, many of these forests have been removed so that only forest patches remain along the rivers upstream of the IND, and only isolated areas are still available within the IND itself. Although there are numerous threats to the wellbeing of resident and migratory birds including human hunting activities and removal of roosts and competition for food, the birds are highly dependent on the mud flats associated with the receding floodwaters of the IND. With reduced flows and inundation areas, and possibly an increase in the rate of flood recession, the migratory birds that depend on the IND are at risk. All of the present and FUT1 development scenarios considered in this project suggest that there will be a low to moderate impact on the bird endpoints considered in the study. Scenarios FUT2–5 will, however, have a high risk to the wellbeing of the bird communities including reductions in food and habitat for birds in the IND. It is important to note that this assessment is based primarily on the effect of altered flows on bird communities based on the existing habitat, roost, and feeding habitats. Additional stressors that may be associated with human population growth and continued habitat removal would probably exacerbate the effects of altered flows, but these have not been considered in this study, as they are not related to e-flows.

### 3.6. Impacts on the Beneficiaries of the Ecosystem Resulting from the Change in Flows

Possible changes to the wellbeing of the ecosystem resulting from flow reductions, as described above, will in turn affect a range of ecosystem services and the human beneficiaries of a functioning healthy ecosystem. Any noticeable reduction in flow will cause a reduction in the abundance of services available to local communities, which will affect their livelihoods. Although the development scenarios considered in this assessment are moderate, some impacts and associated risk to the availability of floodplain vegetation for society, whether it be for grazing of livestock, crops for consumption, or vegetation for construction have been identified.

At risk will be the fisheries that society relies on for supply of much-needed protein and important micronutrients, a sector that is vulnerable to overfishing associated with population growth and commercial export activities. Evaluating the risk of reduced flows on the subsistence fish endpoints (SE2) for all development scenarios (present and future flows 1) shows a change from a zero–low risk towards a low–moderate risk. These results demonstrate that the supply of fish exceeds demand in all riverine risk regions with limited risk of fisheries failure within the IND directly associated with reduced or altered flows. These results do not, however, address potential increases in fishing pressure due to overfishing and or water quality changes, which are already having a serious impact on fisheries. These multiple stressor impacts could be included into this risk assessment during an adaptive management process if required and if data becomes available. In this study, we also considered the risk of changes to the water disease endpoints (SE3) (e.g., malaria, bilharzia, river blindness, dysentery, etc.), associated with the proposed altered flows. Results show a noticeable increase in risk not due to the reductions in flow, but because of changes in the human population size in the area. Findings show that where the abundance of local communities increases, particularly in the IND, the risk of waterborne

disease increases. The abundance of many pest species and vectors of waterborne diseases may be affected by excessively reduced flows, but only a slight response is anticipated for the present and future flow scenarios.

## 4. Discussion

The water resources of the Upper Niger River and Inner Niger Delta are highly variable, dynamic, and driven primarily by the seasonal flows and the associated flooding of the IND. This affects the quality and availability of ecosystem services, habitat availability, and productivity of the ecosystem. Indirect dependents include the biodiversity and key ecosystem processes, many of which are endemic to the area. Recent development of water resources in the Upper Niger Basin that includes dam development, channel modification, and water diversions, as well as land use change and probable climate change, threatens the sustainability of the IND and the ecosystem services it provides. Additional water resource developments that may significantly reduce available flows for the IND may also occur in the future and have been evaluated in this study.

Ecosystems can be described both by the quality and the extent of habitat, both of which will impact their ability to provide ecosystem services to support livelihoods. Floodplains demand, perhaps more so than rivers, that both quality and quantity/extent be considered in setting the e-flow, as any flooding regime that is less than natural will result in reduction in flooding extent and thus the size of the floodplain ecosystem. Most e-flow methodologies embrace the quantity of water as a driver of ecological condition and use the quality of the resulting habitat to ensure that biodiversity is not unacceptably altered and that there is continuation of the ecological structure and function with the resultant flow of ecosystem services to society. These approaches work on the premise that it is possible to reduce the size of the habitat, probably to a threshold size, while maintaining the quality of the ecosystem. What is seldom considered is the extent of habitat that remains available to provide services to society.

A reduction in the extent of a floodplain habitat has two possible outcomes: (1) there is a reduction in the quantity of ecosystem services [42] and thus a proportionate reduction in support to human cultures, economies, livelihoods, and wellbeing [43] and (2) there is a heightened risk to sustainability of the ecosystem, including biodiversity and ecosystem function [44]. The latter will then impact further on the former.

Habitat loss directly affects provisioning and regulating ecosystem services such as food production, raw materials, fresh water and medicinal resources, pollination, pest and disease control, waste-water treatment, erosion prevention and maintenance of soil fertility, regulation of local climate and air quality, etc. The loss of ecosystem services is a clear concern, although studies have shown that the artificial systems that replace the lost habitat can produce more services than before [43,45]. However, the shift to more consumable ecosystem services comes with a risk to services that are perceived to not have an immediate benefit and yet may be of vital importance (e.g., biodiversity). Reduction in floodplain extent could be just such a situation, where the upstream offtake of water for irrigation is perceived to be of greater value than maximal floodplain extent with the many and varied ecosystem services that arise from this. Additionally, such replacements of natural systems with more engineered and artificial ones often leads to commodification of food-producing ecosystem services. It may cause distributional effects where a possible re-distribution of social and environmental costs and benefits of the new system may be inequitable and considered to be socially unfair.

Habitat conversion and loss is associated with species loss as the ability of species to persist in a reduced habitat is threatened. Habitat loss was important enough to garner its own Aichi Target, with target 5 requiring countries to rate the loss of each habitat type and is a target that the world has failed to meet [46]. Sustainable Development Goal target SDG 15.5 also aims to reduce degradation of habitats. Habitat conversion and degradation were regarded as the primary drivers of biodiversity loss in terrestrial and inland water ecosystems [47] leading to permanent loss of valuable ecosystem services [48,49].

Why does a reduced habitat size cause species loss? As habitat is lost, fewer resources are available, there is increasing competition among individuals and decreasing survival and/or reproduction, all leading to a reduction in population size [50], reduced genetic diversity and reproductive success, reduced immigration and dispersal success, increased vulnerability to stochastic events, increased susceptibility to invasive species, and altered interspecies interactions [49,51], all of which increases extinction risk [44]. Biodiversity responses are not necessarily immediate, and different ecological models predict time lags between habitat loss and fragmentation and species extinction [52,53].

The alteration of river flow regimes is claimed to be the most serious threat to ecological sustainability of rivers and their associated floodplain wetlands [54]. The impacts of habitat loss (and climate change) and fragmentation on biodiversity [51,55] are thus considerable. When habitats are lost, then dependent species are also likely to be lost and populations decline [56].

Moss [57] advocated a compromise position for management where joint concern for bio-diversity conservation, ecosystem functioning and resilience, and human livelihoods will provide the most successful long-term basis for freshwater conservation [57]. Implementation of e-flows in the Inner Niger Delta is following this kind of approach.

The size of the IND will be closely related to the inflow, which in turn, will be related to climate as well as upstream developments. Each development scenario will impact on the size of the floodplain. A reduction in flow from the upstream rivers would lead to a reduced wetted area of the IND, i.e., a smaller floodplain, reduced depth of the river channels and ponded areas, and as a result, reduced amount of habitat for plants, mammals, birds, fish, and important invertebrates, which in turn, would all be reduced, leading to reduced population sizes and potentially changing the species dynamics of the whole system. A reduced wetland size would mean encroachment of terrestrial vegetation onto the floodplain, with likely reduced productivity. Additionally, with possibly a higher concentration of people trying to sustain their ecosystem-based livelihoods in those smaller habitats there is a higher risk of over-exploitation. These changes have the potential to impact negatively on the users of the ecosystem.

From this study, we have determined that there is a high likelihood that the floodplain would be reduced in response to the calculated e-flow scenario, while the ecosystem components would be maintained in a satisfactory condition. This model is, however, based on the flow-related drivers of change primarily and does not consider the potential synergistic effects of human pressures on the system. From a social perspective, the larger the floodplain the better, as this would provide the greatest value of services. Rather oblique references to a desired floodplain extent are given in the NBAs sustainable development action plan (Action Plan for the Sustainable Development of the Niger Basin Phase II: Master plan for the development and management 2012), where it is suggested that an 11% reduction in size is the maximum that should be tolerated, as it allows the minimum target flows and agricultural development needs to be met, which would mean the minimum size of the floodplain would be 11,231 $km^2$. This inundation extent is illustrated in Figure 6, however this extent of floodplain size was derived on the basis of willingness of the local people to pay. The e-flows determined in this study, however, would result in a flood extent of 5600 to 5800 $km^2$ (Figure 6), considerably smaller than the more politically derived figure of 11,231 $km^2$.

The PROBFLO outputs include a range of variability in risk to each of the individual social and ecological endpoints as well as the combined social and ecological endpoints. The outputs demonstrate that in 1950 (Reference (REF) scenario), natural flow variability that was only slightly altered from natural, with limited water resources development, resulted in a low to moderate probable risk to all of the endpoints, particularly in the IND. The range of variability in risk even in this near natural state was influenced by the high uncertainty incorporated in the assessment, associated with a lack of biophysical data together with a limited understanding of the flow-ecosystem and flow-ecosystem service relationships.

The results show the high risk to the sustainability of all of the social and ecological endpoints in the IND if scenarios FUT2, FUT3, and FUT4 in particular are implemented and unacceptable risk to the ecosystem if FUT5 scenario is implemented. The implementation of these scenarios will also result in many endpoints becoming unsustainable, and thus, implementation of these scenarios FUT2, FUT3, FUT4, and FUT5, which all represent flows that are lower than the e-flows scenario, is not recommended. The outputs demonstrate that the ecological attributes of the IND considered in this study can be relatively more vulnerable to changes in flows and the flooding regime of the delta in particular, compared with the social endpoints considered. In this case study, we determined that indicator components of the IND ecosystem representing the biodiversity of the system and important ecosystem processes respond rapidly to the reduced flooding regime. The human communities that depend on the ecosystem services of the IND can still obtain numerous products (e.g., fish) and services (e.g., suitable water quality) from the IND when floods that the ecosystem depend on are not provided. Conversely, risk to the social endpoints will be relatively greater for FUT4 scenario compared to the ecological endpoints. The findings of the study demonstrate, however, that excessive reductions in flows and loss of floods will have a significant negative impact on both the ecological and social characteristics of the IND. Should the e-flows be implemented, however, the socio-ecological system will likely be maintained in an acceptable state, however the e-flows can only be achieved if existing water storage facilities maintain flows during dry and or drought periods.

Should an extreme drought occur, similar to that observed in 1984, and be maintained over several years, numerous social and ecological endpoints associated with a healthy IND are likely to be threatened (consider FUT4 scenario). This will result in threats to the wellbeing of the vulnerable human communities that depend on the ecosystem services derived from the IND. The effect of climate change may increase the frequency of droughts and or the duration of droughts that will affects the potential of the system to remain sustainable.

It is important to note that this study and the application of the PROBFLO approach to determine the risks to ecological and social endpoints and to describe the e-flow is low in confidence due to limited data availability. These outputs should only be used for high-level planning studies and for management when the consequences of an error are minor. Confidence in the outputs can be improved by collection of better data, which should be done before these results are used to make any important resource development decisions. It is found that successful implementation of e-flows depends on conducive legislation and regulations, stakeholder participation often led in processes by political champions, sufficient resources and capacities for all involved, and in an adaptive and incremental approach [58].

Implementation of e-flows by implementing the PROBFLO framework in such an adaptive management context will reduce uncertainty over time if the opportunity is taken to collect new data to improve the understanding of the flow–ecosystem relationships. We recommend, therefore, that this framework be implemented in an adaptive management way to improve the confidence of the risk projections over time and to create the right enabling water governance conditions for effective implementation.

This assignment undertook to determine the environmental water requirements or e-flows for the Upper Niger River and the Inner Niger Delta. This was achieved by implementing the ecological e-flow determination and evaluation tool PROBFLO [12]. In this assessment, the socio-ecological consequences of altered flows have been evaluated by assessing the risk of alterations in the volume, duration, and timing of flows, to a number of ecological and social endpoints. Based on the risk posed to these endpoints by each scenario of change, an e-flow was determined that would protect the ecosystem and maintain indicator components at a sustainable level. These e-flows also provide sustainable services to local communities including products for subsistence and limit any abnormal increases in diseases to the vulnerable communities who live in the basin, although this conclusion would be negated if non-flow impacts on the ecosystem increase.

Relative risk outputs for the development scenarios result in low-to-high-risk probabilities for most endpoints. The FUT1 and FUT 5 scenarios in particular are insufficient during dry or low-flow periods with an excessive increase in zero flow possibilities (supplementary information). Although unsuitable during the low-flow or dry periods, sufficient water is available through storage in the basin to meet the e-flows if these scenarios were considered for implementation. Future scenarios FUT2, FUT3, and FUT4, however, have been evaluated, and all of these scenarios will result in significant unacceptable risk to the social and ecological endpoints considered in the study resulting in social and ecological impacts.

The IND is relatively more vulnerable to changes in flows compared to the rivers upstream of the IND. Altered flows will change available habitat in the rivers but critical habitats to maintain the ecosystems and its services upstream of the IND should be available. Note that changes in connectivity between the IND and the rivers upstream will result in significant impacts to the rivers and the IND. Within the IND, altered flows can have significant impacts on the availability and quality of habitats within the IND. These impacts may render the ecosystem and its services unsustainable, resulting in biodiversity losses and disruptions to important processes.

The "wild card" in the future management of the system is climate change. Predictions of what could happen because of climate change are highly uncertain but have been summarized by [59]. Rainfall already shows a substantial decline during the nineteenth and twentieth centuries, reaching a decline of 20% at Timbuktu just below the IND. However, this may be due to regional changes and reflects an alternating trend that has been documented for the region over millennia. Climate change models for the area tend to be highly uncertain, although the preponderance of evidence suggests an increase in temperature and associated decline in river flow. Temperatures may increase by up to 7 °C in the next 80 years, which would have devastating consequences for river flow, water demands and the sustainability of the IND in particular. Given the uncertainy of the predictions for climate change due on data limitations, stakeholders have agreed not to include climate change scenarios in this study as flow-related risk evaluations faced by ecosystems and the community.

## 5. Conclusions

Ensuring that an e-flow for the IND becomes part of mainstream water resource management could help guarantee a sufficiently healthy and functional floodplain ecosystem that provides the ecosystem services required to sustain its flood-dependent society and economy. The aim of this paper has been to demonstrate the contribution that a holistic e-flow assessment can make to floodplain sustainability where the e-flows requirements and consequences of altered flows link to floodplain ecosystems and human livelihoods. In this study, the holistic PROBFLO approach was implemented to determine the e-flow requirements for the UNR and IND and to evaluate the risk of altered flows to contribute to the sustainable management of the IND and the vulnerable ecosystems and people who depend on it.

The outputs of this precautionary desktop e-flow assessment include relatively high percentages of the reference flows (39.32% of the mean annual runoff (MAR) in RR1, 34.43% in RR2, 58.12% in RR3, 40.93% in RR4, 63.35% in RR5, and 58.16% as an inflow from RR5 into the IND) that are required to ensure the sustainability of the ecosystem and its users. Although relatively high, none of the annual hydrological scenarios associated with planned developments (FUT5) will be "worse" than the e-flow requirements proposed for the study. However, when the dry season is considered alone, then the FUT2–5 scenarios do not provide sufficient flow to maintain even the e-flow scenario, suggesting that it will be necessary to release water from the upstream dams during the dry months.

A number of important issues emerge that need to be considered during further planning incorporating the e-flows.

- While the visioning exercise was able to extract from existing policy documents expressions of what is desired for the system, these were not in enough detail to be able to adjust the e-flows accordingly. Ideally, the vision and objectives would rank the balance between use and protection of the ecosystem in terms of risk and would provide benchmarks that would enable the e-flows to

be structured to ensure that the objectives would be met. The greatest uncertainty here, because of a lack of a clear vision and objectives, is that any reduction in flow would mean a reduction in inundation and thus a reduction in services to people. At what point does this become a livelihoods problem? This poses the greatest decision to policy makers in the region.

- The extent of inundation of the floodplain of the IND is a critical issue that is unusual for most e-flows determinations. It was necessary to evaluate all of the endpoints in terms not only of their quality but also the quantity of the service provided. For this study of the IND, a semi-quantitative evaluation of the area of inundation related to natural inundation areas was considered, with associated ecosystem structure and functioning information to determine suitable extent of the flooding for the e-flow determination.

- This study is of the e-flows necessary to sustain the ecosystem and to minimize the risks to the various socio-economic and ecological endpoints. This project does not take into consideration the demand for ecosystem services from communities, which may be greater than the floodplain can provide in any scenario, even if in a reference (near-natural) state. Management of the social consumption of services can have an even greater impact on the state of ecosystem services than management of e-flows.

- Since no site survey was conducted, it was not possible to determine the present condition of the ecosystem and social components. This meant that it was not possible to determine an e-flow to maintain the present situation. As a result, the e-flow that was determined was to describe a lower limit of what would be sustainable. Ideally, this e-flow would not become the objective of resource management but should be seen as a lower limit that should not be exceeded.

- The data that were available to determine the e-flows was only adequate to conduct a low-confidence study. It is important thus that the e-flows that are presented here are recognized as such, as low confidence, and should not be used for high-cost development decisions. A particular weakness of the data was that they were not possible to closely relate the inflows to the extent of floodplain inundation. However, all aspects of the data, with an exception being the hydrology, were weak, and for a higher confidence e-flows assessment, these would need to be strengthened.

- While it appears that there is still abundant water in the system to maintain the ecosystem and users, this is not the case in the dry season where the FUT2–5 scenarios will not provide sufficient water, which would need to be provided from the upstream dams.

- The e-flow requirements are generally less than the development FUT scenario, and thus, it appears that the e-flows will be easy to provide. However, the risks of failure of the various endpoints are higher for the e-flows scenario compared to future development scenarios, driven mostly by vulnerable ecosystem components and estimations of the amount of ecosystem services provided historically. Future decisions on flow reductions should be based on what risk is acceptable to society and a functioning ecosystem.

**Supplementary Materials:** The following are available online at http://www.mdpi.com/2071-1050/12/24/10578/s1.

**Author Contributions:** Conceptualization, C.D. and G.C.O.; methodology, G.C.O. and C.D.; validation, G.C.O., C.D., C.B., F.v.W., and R.S.; formal analysis, G.C.O.; investigation, G.C.O. and C.D.; data curation, G.C.O.; writing—original draft preparation, G.C.O., C.D., C.B., F.v.W., and R.S.; writing—review and editing, C.D., C.B., and F.v.W.; visualization, C.D. and G.C.O.; supervision, C.D.; project administration, C.D.; funding acquisition, C.D. All authors have read and agreed to the published version of the manuscript.

**Funding:** This research was funded by the Dutch Embassy of Mali through a grant to Wetlands International for the BAMGIRE project.

**Acknowledgments:** The International Water Management Institute (IWMI) and the CGIAR Research Program on Water, Land and Ecosystems (WLE) who supported the project. Wetlands International for management of the host BAMGIRE project. Specialists who contributed to the e-flow assessment including Nishadi Eriyagama, Martin Kleynhans, Kate Rowntree, Mark Graham, Vere Ross-Gillespie, James MacKenzie, Erik Klop, and Eddy Wymenga of Altenburg and Wymenga and Everisto Mapedza; Stephan Liersch of the Potsdam Institute for Climate Impact Research (PIK), Md Mominul Haquea and Ousmane Seidou of University of Ottawa, and Stijn Schep of Wolfs Company who all provided data.

**Conflicts of Interest:** The authors declare no conflict of interest. The funders had no role in the design of the study; in the collection, analyses, or interpretation of data; in the writing of the manuscript, or in the decision to publish the results.

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
