# Peer review of "Sustainable Floodplains: Linking E-Flows to Floodplain Management, Ecosystems, and Livelihoods in the Sahel of North Africa"

_sustainability, doi:10.3390/su122410578_

Round 1

Reviewer 1 Report

The paper topic is interesting as well as the model and the approch used to present the results. I have only minor comments to improve the readability of the paper:

  • it would be useful to provide in the study area description the climate change profile of the region (if there are in the literature climate change scenarios for the region)
  • in the methodology section, please pay attention to describe more accurately the data you have used: e.g. which is the data source, for which time period you make the analysis - for all kind of data. You mentioned explicitly only flow data, what did you use to characterised socio-economic parameters or habitat data etc. You discuss a lot in the paper about data availability and quality, so a clear section referring to data should be made available in the methodological section
  • the paper should have a well-defined conclusion section, separated from the discussion
  • please carefully check the manuscript as some spelling corrections are necessary: e.g. page 2, line 62 "improived" should be "improved", line 65 correct life.h.
  • line 72 The should be on the next line
  • line 81 correct "aa"
  • line 208 - evaluated work is repeated. Better use "has been analysed"
  • line 214 - replace manuscript with paper

Author Response

Thank you for your useful comments. We have addressed all of your requests.

Reviewer general comment: The paper topic is interesting as well as the model and the approach used to present the results. I have only minor comments to improve the readability of the paper:

Comment 1: it would be useful to provide in the study area description the climate change profile of the region (if there are in the literature climate change scenarios for the region).

Response 1: Addressed with the inclusion of multiple inclusions of information describing climate change issues in the region. Including: “Exacerbating this, climate change is set to compound these changes in the natural flooding dynamics, a decline in per capita water availability in Mali by 77% by 2080 compared to 2000 (PIK, 2020).”

Comment 2: in the methodology section, please pay attention to describe more accurately the data you have used: e.g. which is the data source, for which time period you make the analysis - for all kind of data. You mentioned explicitly only flow data, what did you use to characterised socio-economic parameters or habitat data etc. You discuss a lot in the paper about data availability and quality, so a clear section referring to data should be made available in the methodological section.

Response 1: We have improved our reference to the data used and referenced in the supplementary information by including: “Data used in the risk assessment was generated through extensive specialist reviews of available information describing the socio-ecological systems of the UNR&IND, and solicitations from regional Sahealian and Sub-Saharan Africa information presented in IWMI (2018), and reviewed and referenced in the supplementary information.”     

Comment 3: the paper should have a well-defined conclusion section, separated from the discussion

Response 1: This section has been reworked to addess this comment. Thank you.

Comment 4: please carefully check the manuscript as some spelling corrections are necessary: e.g. page 2, line 62 "improived" should be "improved", line 65 correct life.h.

Response 1: the language of the MS has been carefully reviewed and improved.  

Comment 5: line 72 The should be on the next line

Response 1: addressed as requested.

Comment 6: line 81 correct "aa"

Response 1: corrected

Comment 7: line 208 - evaluated work is repeated. Better use "has been analysed"

Response 1: corrected

Comment 8: line 214 - replace manuscript with paper

Response 1: corrected

Reviewer 2 Report

Comments for the authors:

The manuscripts “Sustainable Floodplains: Linking E-Flows to Floodplain Management, Ecosystems and Livelihoods in the Sahel of North Africa”, which aims to demonstrate the contribution of regional environmental flow using PROBFLO to establish floodplain sustainability in Inner Niger Delta in Mali, fits within the journal’s scope and is generally well-structured. It provides some background of previous works. However, there are several major points which need to be addressed and significantly improved before proceeding further. The most important issue that was apparent in the manuscript was many sentences with grammatical or punctuation errors. These include Line 24 where the aim of this manuscript is described, Line 52 where the SSA is not defined, Line 72 where there is a typo at the end of the sentence. Line 81 and many other examples. Also, there are several sentences with unnecessary italic fonts which is not professional. I highly suggest the authors to ask a native English speaker or an English institute to check the text of the manuscript critically. Also, it would be ideal if you define the e-flow in the abstract before describing its contribution to flood plain management. I also suggest to add the research questions you are trying to address as well as the fundamental contributions of this work specifically at the end of Section 1. Also, I suggest to split the Discussion Section and add a Conclusion Section which the major findings of this research are clearly described, preferably point by point.

Author Response

Thank you for your useful comments. We have addressed all of your requests.

Reviewer 2 comments: The manuscripts “Sustainable Floodplains: Linking E-Flows to Floodplain Management, Ecosystems and Livelihoods in the Sahel of North Africa”, which aims to demonstrate the contribution of regional environmental flow using PROBFLO to establish floodplain sustainability in Inner Niger Delta in Mali, fits within the journal’s scope and is generally well-structured. It provides some background of previous works. However, there are several major points which need to be addressed and significantly improved before proceeding further.

Comment 1: The most important issue that was apparent in the manuscript was many sentences with grammatical or punctuation errors.

Response 1: The MS has been carefully reviewed.

Comment 2: These include Line 24 where the aim of this manuscript is described,

Response 2: amended as requested as follows: “This paper aims to demonstrate the contribution that holistic regional e-flow assessment using the PROBFLO approach has to achieving floodplain sustainability. This can be achieved through the determining the e-flow requirements to maintain critical requirements of the ecosystems and associated services used by local vulnerable human communities for subsistence and describing the socio-ecological consequences of altered flows. These outcomes can contribute to the management of the IND.”

Comment 3: Line 52 where the SSA is not defined,

Response 3: Corrected as requested.

Comment 4: Line 72 where there is a typo at the end of the sentence.

Response 4:  Corrected as requested.

Comment 5: Line 81 and many other examples.

Response 5: Corrected as requested.

Comment 6: Also, there are several sentences with unnecessary italic fonts which is not professional. I highly suggest the authors to ask a native English speaker or an English institute to check the text of the manuscript critically.

Response 6: Amended as requested. Only non-english language terms used in the MS have been italicised.

Comment 7: Also, it would be ideal if you define the e-flow in the abstract before describing its contribution to flood plain management.

Response 7: included as recommended with “Environmental flows (e-flows) include the quantity and timing of flows or water levels needed to meet the sustainable requirements of freshwater and estuarine ecosystems.”

Comment 8: I also suggest to add the research questions you are trying to address as well as the fundamental contributions of this work specifically at the end of Section 1.

Response 8: This has been include in the MS generally throughout following the requirements of reviewer 1.  A specific research question has been included as follows: “The research question for the study queries if suitable, holistic e-flows can be established for the UNR&IND on appropriate spatial scales that address social and ecological features and values of the system that will contribute to sustainable floodplain management for the people and environment of the IND.” 

Comment 9: Also, I suggest to split the Discussion Section and add a Conclusion Section which the major findings of this research are clearly described, preferably point by point.

Response 9: Amended as requested.

Round 2

Reviewer 2 Report

I believe the authors have addressed most of my comments properly and I think the manuscript can be accepted in the current format. However, I suggest the authors to double-check the English of the manuscript carefully.